# Implications of Endogenous Retroelements in the Etiopathogenesis of Systemic Lupus Erythematosus

**DOI:** 10.3390/jcm10040856

**Published:** 2021-02-19

**Authors:** Kennedy C. Ukadike, Tomas Mustelin

**Affiliations:** Division of Rheumatology, Department of Medicine, University of Washington School of Medicine, 750 Republican Street, Seattle, WA 98109, USA; kukadike@uw.edu

**Keywords:** systemic lupus erythematosus, retroelements, L1, LINE-1, reverse transcriptase, type I interferons, autoimmunity

## Abstract

Systemic lupus erythematosus (SLE) is a heterogeneous autoimmune disease. While its etiology remains elusive, current understanding suggests a multifactorial process with contributions by genetic, immunologic, hormonal, and environmental factors. A hypothesis that combines several of these factors proposes that genomic elements, the L1 retrotransposons, are instrumental in SLE pathogenesis. L1 retroelements are transcriptionally activated in SLE and produce two proteins, ORF1p and ORF2p, which are immunogenic and can drive type I interferon (IFN) production by producing DNA species that activate cytosolic DNA sensors. In addition, these two proteins reside in RNA-rich macromolecular assemblies that also contain well-known SLE autoantigens like Ro60. We surmise that cells expressing L1 will exhibit all the hallmarks of cells infected by a virus, resulting in a cellular and humoral immune response similar to those in chronic viral infections. However, unlike exogenous viruses, L1 retroelements cannot be eliminated from the host genome. Hence, dysregulated L1 will cause a chronic, but perhaps episodic, challenge for the immune system. The clinical and immunological features of SLE can be at least partly explained by this model. Here we review the support for, and the gaps in, this hypothesis of SLE and its potential for new diagnostic, prognostic, and therapeutic options in SLE.

## 1. Introduction

Systemic lupus erythematosus (SLE) is a varied and often debilitating autoimmune disease that affects at least 5 million people worldwide, and women more than men with a striking gender bias of 9:1. The precise etiology of SLE remains elusive despite many decades of research to better understand it. Current knowledge suggests a multifactorial etiology with contributions from genetic, immunologic, hormonal, and environmental factors [1,2]. Even at that, the exact extent to which each of these factors contribute to SLE pathogenesis is not known. While we focus here on a specific emerging mechanism that combines genomic/genetic and immunologic factors, with hormonal and environmental contributions, we wish to first place it in the context of the broader genetic associations of SLE.

Genome-wide association studies have identified many genes with polymorphisms and copy number variants that are associated with SLE [3,4,5,6,7]. The most significant associations are found in the major histocompatibility complex II (MHC II), which include alleles of *HLA-DR2*, *HLA-DR3*, and *HLA-DQ2* [8,9,10]. Deficiencies of the complement components C1q [11], C2, C4A, and C4B, which confer an even higher risk for SLE, are relatively rare [12]. Similarly, rare polymorphisms or mutations in DNases *TREX1* [13] and *DNASE1* [14] also confer significant risk of SLE. Deletion of *trex1* in mice results in accumulation of single-stranded DNA derived from reverse transcription of retroelement RNA, elevated type I interferon production, and severe autoimmunity [15]. In humans, loss-of-function mutations in *DNASE1L3* also result in a SLE-like disease [16]. This gene encodes for an active DNase that is secreted by innate immune cells to degrade chromatin released passively (apoptosis and necrosis) or actively (NETosis) from dying cells. Together, these genes imply a pathogenic role of cytosolic DNA originating from retroelements, and the importance of effective clearance of DNA in immune complexes and cellular debris.

In agreement with this notion, several genes with a role in IFN signaling, such as *IRF5*, *IRAK1*, *STAT4*, *SPP1*, *TNFAIP3*, and *PTPN22*, also have SLE-predisposing variants, which are associated with high levels of type I IFNs and increased expression of IFN-inducible genes [17,18,19,20,21]. Polymorphisms in genes involved upstream of IFNs, such as *IFIH1* [22] and *TLR7* [23], have also been documented. Other genes implicated in the adaptive immune system, including *PTPN22*, *PDCD1* (encodes PD-1) [24], *BANK-1* [25], *BLK*, *LYN*, and *TNFRSF4* (OX40L), indicate that the threshold for activation of B and T cells is important in SLE [26,27,28,29]. The MHC association also supports this notion. Unlike the rare complement deficiencies and DNase mutations, these gene polymorphisms individually confer a very modest risk (odds ratio <2) for SLE, suggesting that they are not directly causative, but in aggregate increase the susceptibility to SLE, presumably in combination with the absence of protective gene variants [30,31], genomic hypomethylation, altered epigenetic control, changes in microRNAs (miRNAs) [32,33,34,35,36], and the presence of environmental or endogenous triggers [34,35,36].

In accordance with the genetics of SLE summarized above, we focus in this review on an emerging concept that is well compatible with the genetic associations, namely the notion that endogenous virus-like sequences may play a part in the pathogenesis of SLE and other related diseases [37,38,39,40]. These genomic sequences are either remnants of exogenous retroviruses that infected our ancestors millions of years ago [40,41,42], or ancient descendants of retroviruses that retained the ability to embed and replicate within the germline genome to become extremely abundant throughout the human genome [40,43]. Although the vast majority of all these sequences are now inactive due to mutations and truncations, a number of them are still more or less intact and able to create extra-chromosomal DNA, trigger type I IFNs, and provoke an antiviral type of immune response. The biology of these retroelements and the evidence for their involvement in SLE are discussed here.

## 2. Transposable Elements in the Human Genome

Colloquially known as “jumping genes” or “parasitic DNA” [44], transposable elements (or transposons) are genomic DNA sequences that have the ability to move within the genome, thereby altering its organization, incrementally increasing its size, and creating duplications and redundancy [45]. There are two broad classes of transposons: Class I transposons, also known as retrotransposons, and class II or DNA transposons [46]. The former propagate using a “copy-and-paste” mechanism that consists of a reverse transcriptase (RT) that uses its own RNA transcript as a template to generate a cDNA copy, which is inserted into the genome. The latter move by a “cut-and-paste” mechanism by their encoded transposase enzyme. To the best of our knowledge, only class I transposons have been implicated in the autoimmune disease and will be discussed further here.

To illustrate the sheer volume of retrotransposons in our genome, compared to all the exons of our 20,000 genes, which occupy approximately 1% of our 3-billion base-pair genome, the retroelements occupy close to 50% of it [44,47]. There are over 3 million retroelements in our genome [48]. They fall into three categories: the over 440,000 long terminal repeat (LTR) retrotransposons, also known as human endogenous retroviruses (HERVs), the 800,000 autonomous non-LTR retrotransposons termed long interspersed nuclear elements (LINEs), and the 1,500,000 copies of the short interspersed nuclear elements (SINEs), which are non-autonomous and include over 1 million Alu elements [49] (Figure 1).

Before delving into the immunological impacts of retroelements, it should be stated that the retrotransposition mechanism itself can cause genomic damage and result in human disease [50]. New retrotransposon insertions in or near exonic genes can result in altered transcription [51], disrupted mRNA splicing, premature termination of translation, and loss of protein expression or function. Besides sporadic genetic diseases [52] caused by new retrotranspositions, this biology is accelerated in malignant cells [53] and is a major contributor to the activation of oncogenes [54], the inactivation of tumor suppressors [55,56], and larger chromosomal abnormalities [50,57,58,59]. Retroelements are also abundant around chromosome fragile sites, such as FRA3B on chromosome 3p14 and FRA16D on chromosome 16q23 [60,61].

### 2.1. HERVs

The HERVs are the very definition of autonomous retrotransposons in that they resulted from germline infections by exogenous retroviruses that upon cell entry reverse-transcribed their RNA genome and inserted it into the host cell genome. The resulting HERVs were subsequently passed on to offspring in a Mendelian fashion and most of them exist in all now living humans [62]. Transcription of such newly formed HERVs result in a polycistronic transcript that, after splicing, encodes for all the proteins necessary for the formation of new infectious virions [63]. However, because HERVs are not under positive selection pressures (but rather the opposite), they accumulate random mutations, deletions, insertions, recombinations, and other genetic alterations over evolutionary time [62]. The modern human genome does not appear to contain any fully intact and functional HERVs anymore [62,64,65], but there still are about a dozen HERVs that encode for proteins that have some, or all, of their original functions [63,64,65,66,67]. Some of the youngest (=most recently incorporated) HERVs can still form virions [68], even though they lack measurable infectivity.

The HERVs in our genome belong to three classes: gammaretroviruses (class I), betaretroviruses (class II), and spumaretroviruses (class III) [69]. The published literature proposes various roles for class I (HERV-E, and to a lesser extent -W, and -H) and class II (HERV-K) HERVs in autoimmune diseases [70,71,72,73]. A common denominator among these papers is the idea that their transcriptional upregulation will trigger various aspects of an antiviral immune response, including autoantibodies against retroviral proteins [74,75,76,77]. A popular suggestion is that HERV proteins may trigger autoimmunity by molecular mimicry [70,78] through accidental similarities between these proteins and other self-proteins. However, we believe that an immune response against HERV proteins already constitutes “autoimmunity” whether any cross-reactivity exists with proteins encoded by exonic genes, or not.

It should also be kept in mind that even HERVs that have lost their ability to encode for proteins often still possess their strong transactivating long-terminal repeats (LTRs) [79], which can influence the transcription of nearby protein-coding genes [51]. This appears to be a driver of altered gene expression in cancer [80,81], where demethylated LTRs can respond to transcription factors, including those activated by sex hormones. Demethylation of LTR sequences reportedly upregulates HERV expression also in autoimmune diseases like SLE [82,83]. An example of this is the influence on RAB4 gene expression exerted by the demethylated LTR of a truncated class I HERV element, termed HRES-1 [78]. RAB4, in turn, downregulates surface CD4 expression, which together with the immunogenic 28-kDa Gag protein of HRES-1 can contribute to the self-reactivity of T and B cells in SLE [78]. Interestingly, polymorphisms in the HRES-I LTR are associated with SLE [84].

### 2.2. L1 Retrotransposons

Intact and functional LINE retrotransposons are also autonomous in that they encode all the components needed for their own retrotransposition [44,85,86]. This machinery is also responsible for the retrotranspositions of the non-autonomous retrotransposons [87], and for creating all our pseudogenes [44]. Research has focused primarily on LINE-1 (or L1), which not only are abundant, but also include members that have retained all or some of their biological functions. In contrast, the LINE-2 and LINE-3 groups, although still prevalent, are all inactive, but can serve as templates for regulatory RNA species [88].

As depicted in Figure 2, the L1 transcript is bicistronic and encodes for two proteins, the 40-kDa RNA-binding protein ORF1p and the 149-kDa endonuclease [89] and reverse transcriptase ORF2p [90], which assemble in approximately a 20:1 stoichiometry into complexes with high affinity for RNA, particularly L1 mRNA, but also Alu RNA and other small RNAs [85]. To execute retrotransposition, these ORF1p/ORF2p/RNA translocate to the nucleus, where the endonuclease activity of ORF2p cuts the genome at a poly-dT tract, allowing the poly-A tail of the L1 transcript to align, enabling the reverse transcriptase activity of ORF2p to synthesize a cDNA copy of the associated RNA, followed by DNA repair [85] (Figure 2). As a result, the genome now has a new 6-kb L1 element identical to the one that created it. New Alu elements and pseudogenes are generated by the exact same mechanism [44].

While there is presently no conclusive evidence that retrotransposition of L1 plays any role in autoimmunity (and no evidence that it does not), there are several other aspects of L1 biology that make these elements prime suspects in the pathogenesis of SLE and related autoimmune diseases characterized by elevated type I IFNs.

### 2.3. Non-Autonomous Retroelements

The enormous abundance of Alu elements with over one million copies throughout our genome, all generated by the L1 retrotransposition machinery, bears witness to the period of very active genome remodeling during hominid evolution. Alu elements are found abundantly within introns and in regulatory regions of genes and in intergenic space. The generation of new Alu and SVA elements is still ongoing and can result in positive or negative changes in the transcriptional control of genes. As such, this mechanism can contribute to human disease, conceivably including autoimmune diseases like lupus. An example of this was the discovery of an Alu insertion into an intron of the FAS/CD95 gene, which resulted in mis-splicing of its transcript, loss of functional FAS protein, and lymphoproliferative disease [91]. Alu transcripts also have the potential to form double-stranded structures, which can be recognized by RNA sensors to induce type I IFNs [92]. This danger is normally reduced by adenosine-to-inosine editing by the ADAR1 enzyme [93], the loss of which causes the interferonopathy Aicardi–Goutières syndrome, discussed in Section 3.2. This RNA editing also appears to be defective in patients with multiple sclerosis [94].

Alu elements have also gained interest in lupus research due to the association of Alu-derived RNA with Ro60 [95,96,97], a well-recognized SLE autoantigen. In a 2015 paper [97], immune complexes formed by anti-Ro60 autoantibodies where isolated from SLE patients and the bound RNA sequenced, revealing that much of it was Alu- and L1-derived. We will discuss the protein and RNA complexes that contain Ro60, known as stress granules, more below.

## 3. How L1 Retrotransposons May Trigger IFN-Positive SLE

There are several reasons to ask whether L1 retrotransposons play an important role in the pathogenesis and flares of SLE. Increased L1 transcripts and ORF1p protein have been detected in kidney biopsies from patients with lupus nephritis and in salivary gland biopsies from Sjögren’s syndrome patients [98]. In healthy individuals, L1 transcripts are low or undetectable, but can be induced by demethylating drugs like 5-aza-deoxycytosine [99], including those known to cause drug-induced lupus [100,101], e.g., hydralazine and procainamide. Reduced methylation of the 5′ regulatory (“promoter”) region of L1 has been reported in both adult and pediatric lupus patients [102]. UV light, a well-known trigger of lupus flares [103,104], also causes DNA demethylation, in addition to causing direct DNA damage and cell death at higher exposures. L1 expression also responds to other environmental and microbial factors [105,106].

Essentially all patients with SLE have IgG autoantibodies against ORF1p [107,108], which correlate with disease activity measured by the SLE disease activity index (SLEDAI), the presence of lupus nephritis, complement consumption, increased anti-dsDNA, and higher type I IFN activity [107]. Importantly, there anti-ORF1p autoantibodies do not represent anti-DNA reactivity, as free dsDNA did not compete (while free ORF1p did), DNase treatment did not affect them (while it eliminated anti-dsDNA reactivity), and ORF1p was recognized even when mixed with whole cell lysates. Presumably related to this finding, ORF1p and ORF2p reside in cells in macromolecular assemblies referred to as “stress-granules” [109], which are rich in RNA and RNA-binding proteins, including Ro60 and other SLE autoantigens [110].

Importantly, L1 expression has been shown to induce type I IFNs [111,112,113], which are a hallmark of SLE [114,115,116,117,118]. This can reportedly occur by two different mechanism [111,112,113,119], which are not mutually exclusive: (i) cytosolic DNA generated by reverse transcription by ORF2p activates DNA sensors [111], such as cyclic guanosine adenosine monophosphate synthase (cGAS), which through the stimulator of interferon genes (STING) adapter protein [120] activates the TBK1 protein kinase [121], which phosphorylates the IRF3 transcription factor leading to type 1 IFN production. Indeed, cGAS activation was documented in some 17% of SLE in a recent study [122]; (ii) double-stranded RNA species [113], perhaps related to bidirectional L1 promoter activity, activates RNA sensors that initiate the same kinase-transcription factor pathway to type I IFNs. While this second pathway is not restricted to L1 transcripts, either, or both, of these mechanisms can explain the elevated expression of IFN-inducible genes, referred to as the “IFN signature” [116,123] in SLE and related autoimmune diseases, such as idiopathic inflammatory myopathies and primary Sjögren’s syndrome [124].

Taking all these observations together, it appears that L1 elements with intact ORF1 and ORF2 are derepressed by reduced DNA methylation (and other epigenetic mechanisms that depend on it) and, therefore, transcribed at elevated levels compared to healthy individuals. Indeed, decreased DNA methylation has been documented in SLE, including specifically in the 5′ regulatory regions of L1 [102]. However, there are also reports that L1 methylation is not altered, but one has to keep in mind that such measures are a composite of numerous L1 elements and does not necessarily represent the relatively small number of L1 loci that are transcriptionally activated in SLE. The epigenetic regulation of L1 elements also varies between cell types. Even different immune cell lineages have distinct patterns of active L1 elements (our unpublished observation).

Translation of these elevated L1 transcripts leads to accumulation of ORF1p and ORF2p in stress granules [109], which, because they contain immunogenic ORF1p protein and lots of RNA, seem to be of special interest to the immune system in SLE patients. We surmise that cells expressing L1, containing triggered DNA and/or RNA sensors, and producing type I IFNs, will appear virally infected to the host immune system and drive a chronic and/or episodic systemic inflammation, which will escalate every time L1 transcription increases. Since the culprit L1 elements cannot be eradicated from the genome, the frustrated immune response will increase in magnitude with time and eventually be diagnosed as SLE.

This model (Figure 3) illustrates how L1 may contribute to many of the well-recognized aspects of SLE: its long prediagnosis development [125] and gradual presentation, its unpredictable and relapsing/remitting nature, the high type I interferons, its sensitivity to demethylating drugs and UV, and the focus of the autoimmune response towards nucleic acids and proteins associated with them. These features also explain the typical symptoms of SLE, such as fever, fatigue, arthralgias, and the multitude of organ manifestations related to the accumulation of immune complexes.

### 3.1. HERVs and Other Non-L1 Retrotransposons in SLE

Elevated expression [67,126] of many HERVs and autoantibodies against HERV-K and HERV-E Gag and Env proteins [40,72,74,75,76] have been reported in SLE [127] and other autoimmune diseases [71]. The broader genomic hypomethylation observed in SLE may well explain the upregulation of HERV transcription, but since most HERVs have lost their ability to encode full-length retroviral proteins, only a few of these transcripts are capable of supporting autoantibody production. The resulting autoantibodies may synergize with anti-L1 immunity, for example, in the formation of immune complexes that drive tissue inflammation and organ damage. HERVs with an intact pol gene, encoding for their reverse transcriptase, can, in principle, produce DNA species that trigger DNA sensors like cGAS or ZBP-1 to induce type I IFN production. However, the retroviral life-cycle involves a protected reverse transcription of the RNA genome only upon cell entry and in the confines of the nucleocapsid [128,129]. Hence, HERVs are not likely to generate pathogenic DNA in SLE, but they may well generate double-stranded RNA transcripts that can trigger RNA sensors.

### 3.2. Are Defenses Against L1 and HERVs Defective in SLE?

Although many components of the model presented above are well documented, it still contains significant gaps. Why does L1 become hypomethylated in individuals who develop SLE? Why is ORF1p so immunogenic? What prevents this from occurring in healthy individuals?

Since majority of people never develop SLE, there must be effective mechanisms to counteract the biology of L1 and HERVs to prevent their deleterious effects on our health. Indeed, numerous defenses exist against all retrotransposons [130,131], many discovered during research into the infectivity of human immunodeficiency virus (HIV). These defenses operate at every step of the life-cycle of retrotransposons and HERVs, and exogenous retroviruses. Some of these defense mechanisms also operate to combat other exogenous RNA and DNA viruses.

Epigenetic regulation is a fundamental mechanism employed by cells to silence genes whose actions are either not needed or are potentially deleterious [132]. This mechanism of transcriptional repression operates on L1 [132] and HERVs and is initiated by DNA methyltransferase 1 (DNMT1) [133], which methylates the 5-position of cytosine in genomic CpG islands, attracting several silencing factors such as the human silencing hub (HUSH) complex [134] and histone modifiers [135] to effectively suppress transcription. Next, RNA interference and silencing activities of small interfering RNAs (siRNAs), miRNAs, and Piwi-interacting RNAs (piRNAs) act to prevent retrotransposon mRNA translation [136]. Of these, the piRNA system is particularly important for protecting the integrity of the germline genome against retrotransposons [137,138].

Hypomethylation of the genome [139] and specific hypomethylation of L1 elements and HERVs have been documented in SLE [140,141] and Sjögren’s syndrome [139,141]. The epigenetic mechanisms of L1 repression may also be influenced by environmental factors [142,143]. It is intriguing that drugs known to cause drug-induced lupus, such as hydralazine and procainamide [144,145], and UV light exposure (a well-known trigger of lupus flares [104]), are demethylating agents [146] and increase L1 and HERV expression.

In concert with the above mechanisms, the cytosolic DNase TREX1 [147] and the heterotrimeric RNaseH2 enzyme [148] act to remove cytosolic DNA [15] and RNA species, respectively. Both enzymes are particularly active against DNA:RNA hybrids [149], the intermediate stage of reverse transcription. Indeed, loss of TREX1 results in accumulation of L1-catalyzed DNA in cytosolic granules [149,150]. The importance of these nucleic acid degrading enzymes is perhaps best illustrated by their loss-of-function mutations [151] in Aicardi–Goutières syndrome (AGS) a devastating disease characterized by constitutively high production of type I IFNs, neurologic deficits due to IFN toxicity, and autoimmunity with all the hallmarks and autoantibodies of SLE [152]. L1 expression is high in AGS [153] and type I IFN production can be reduced by administering reverse transcriptase inhibitors that are active against ORF2p [154]. The form of SLE caused by TREX1 mutations [13] likely involves the same overproduction of ORF2p-generated DNA.

The function of retrotransposon proteins is also targeted by defense mechanisms, such as translational inhibition by the ATP-dependent RNA helicase Moloney leukemia virus 10 (MOV10) protein [155,156,157], which coexists with ORF1p in stress granules [110]. Exactly how MOV10 works is not well understood. Another L1-associated protein identified by proteomics [110,158] is zinc finger CCHC domain-containing 3 (ZCCHC3), a cofactor for both DNA and RNA sensors [159,160]. The *SAMHD1* gene, loss-of-function mutations of which also lead to AGS [161], encodes a phosphohydrolase that dephosphorylates the deoxy-nucleotide triphosphates required for reverse transcription. In addition, the retrotransposition process is directly disrupted by mutation-inducing members of the apolipoprotein B mRNA editing catalytic polypeptide-like 3 (APOBEC3) family of enzymes [162,163], which deaminate cytosines to uracil, and adenosine deaminase of RNA 1 (ADAR1) [93], which deaminates adenosines to inosine. As a result of these mechanisms, the majority of all retrotranspositions result in mutated and severely 5′ truncated new copies (reverse transcription starts in the 3′ end). Most importantly, these mechanisms counteract the production of IFN-inducing DNA and other aspects of L1 biology that can lead to immune activation. Future work will determine if any of these mechanisms are defective in SLE.

### 3.3. Subsets of SLE with Distinct Mechanisms

The therapeutic options for the management of SLE are limited and often fail to control the disease without unacceptable adverse events. Numerous candidate drugs have failed in clinical trials, for reasons that likely include its molecular heterogeneity and the inaccuracy of tools to assess disease activity. It is quite possible that no single drug will be effective and safe in all SLE patients, but that the precision medicine concept of “the right medicine for the right patient” is particularly relevant in SLE.

Based on biochemical and available clinical trial data, we proposed recently that SLE consists of at least four distinct molecular “endotypes” [164]. The first of these is the IFN-independent form of SLE, “SLE1”, defined as the patients who meet the diagnostic criteria for SLE, but consistently lack an IFN signature, i.e., IFN-induced genes are expressed at normal low levels. The remaining three endotypes are characterized by a positive IFN signature, but differ in which nucleic acid sensors have been activated and, consequently, which isotypes of type I IFNs are overproduced.

We define SLE2 as the form in which extracellular immune complexes that contain nucleic acids (e.g., L1-containing stress granules) activate endosomal toll-like receptors (TLRs) 3, 7, 8, or 9 to induce type I IFN production [165]. Due to the predominant expression of TLRs in immune cells, particularly plasmacytoid [166], but also myeloid dendritic cells, macrophages, monocytes, and B cells, the spectrum of induced IFNs include numerous isotypes of IFNα with lesser contributions by IFNβ and type III IFNs [123]. This form of SLE was previously thought to be the main form [167], but the failures in phase 2 clinical trials of multiple TLR7/9 antagonists and antibodies like rontalizumab and sifalimumab that effectively neutralize IFNα, indicate that only 10% or less of SLE patients have SLE2. Most telling, the elevated IFN-inducible genes in the blood of treated patients only declined marginally in patients treated with sifalimumab.

SLE3 is an IFNβ-predominant endotype with activated cytosolic DNA and/or RNA sensors, representing the two alternative mechanisms by which L1 can drive type I IFNs. This biology can occur in any cell type that expresses L1 and/or produces pathogenic double-stranded RNA and this is also how exogenous DNA or RNA viruses initiate an antiviral immune response.

We consider it plausible that SLE typically starts as a pure SLE3 endotype, but that the immune response eventually escalates to a stage where circulating immune complexes with L1-containing, or other, RNA-rich particles accumulate and begin to trigger TLRs on immune cells, i.e., inducing the SLE2 endotype mechanism for type I IFN production. We designated this overlap as SLE4, in which all type I IFNs are at play and both cytosolic and endosomal nucleic acid sensors are active. We estimated that SLE1 represents 10–30%, SLE2 less than 10%, and SLE3 and SLE4 together 60–80% of all SLE patients.

Support for this molecular classification comes from clinical trials with drugs that target IFNs, such as rontalizumab (anti-IFNα), sifalimumab (anti-IFNα), and anifrolumab (antitype I IFN receptor) [168,169,170], bearing in mind that average outcomes are not as illuminating as a more detailed responder vs. non-responder assessment. Indeed, it is likely that many clinical trial failures in SLE, e.g., with TLR7 antagonists, are the results of too few patients of the responding endotype. In this scenario, the patients with non-responding endotypes diluted out the therapeutic effects beyond the statistical analysis of the entire intent-to-treat cohort.

### 3.4. L1- and HERV-Related Biomarkers

Whether the above classification is relevant or not, SLE is clearly a heterogeneous disease in its clinical manifestations and response to therapy [1,2]. Many tools have been developed and revised over the years to help guide the diagnosis and management of patients with SLE, and to measure therapeutic effects of drugs during clinical trials. They include various high-sensitivity and high-specificity clinical- and laboratory-based classification criteria (e.g., SLICC criteria) and disease activity indices (e.g., SLEDAI). Despite all these tools, however, the management of SLE, especially in severe disease states, remains one of the biggest challenges in rheumatology. There is often discordance between laboratory evidence of immunologic activity and clinical evidence of disease activity. New diagnostic tools or biomarkers might help narrow the gap.

As we recently demonstrated, the titers of IgG autoantibodies against L1 ORF1p correlate significantly with disease phenotypes, SLEDAI, markers of disease activity, and IFN score [107]. These autoantibodies could conceivably aid in the diagnosis and prognosis of the disease, perhaps guiding which endotype of SLE an individual patient has and, hence, which treatment regimen might be most effective. High titers of anti-ORF1p autoantibodies may also help identify patients who progress to end organ damage, such as lupus nephritis, and may benefit from earlier optimization of their treatment. This would need to be rigorously tested in prospective clinical studies.

Another set of biomarkers would be tests for the activation of the DNA and RNA sensors. Quantitation of the unique second messenger that cGAS produces, cyclic-guanosine adenosine-2,3-monophosphate (cGAMP) by mass spectrometry is probably too cumbersome for use in clinical practice, but newer high-sensitivity ELISAs are under development. For example, it would make sense to consider cGAS inhibitors specifically in those patients that are positive for cGAMP. Another biomarker to reveal the activation of the RNA sensors could be useful. When triggered, these sensors cause the oligomerization of the mitochondrial antiviral signaling (MAVS) adaptor protein, a response that is readily detectable on non-denaturing gels as an ultrahigh-molecular weight species [171]. Representative individual isotypes of the 17 type I IFNs can be quantitated by the ultra-sensitive single-molecule array (SIMOA) platform [172].

### 3.5. Novel Therapeutic Opportunities Related to L1

New effective and safe drugs are urgently needed for SLE. It stands to reason that drugs that selectively interfere with the molecular pathways that drive SLE, rather than broadly suppress the immune system, would be both more effective and better tolerated than current treatments. The L1 mechanisms we discussed above offer a new option, at least for the SLE3 and SLE4 endotypes, namely the inhibition of ORF2p-catalyzed reverse transcription, which is upstream of type I IFN production, and all the other biological responses induced by activated DNA sensors, such as the upregulation of MHC and costimulatory molecules. Of the FDA-approved reverse transcriptase inhibitors used for the treatment of HIV, some nucleoside RTIs (NRTIs) are equally or near-equally potent on ORF2p as on HIV RT, while others, including the non-nucleoside reverse transcriptase inhibitors, are not. Studies in Trex1-deficient mice, which suffer from severe autoimmune myocarditis and high type I IFNs similar to AGS, have shown that these mice can be rescued by treatment with a three-drug NRTI combination (emtricitabine, tenofovir, and nevirapine). Even more striking, patients with AGS treated with an FDA-approved three-drug NRTI regimen (abacavir, lamivudine, and zidovudine) showed marked reduction in the levels of IFNα proteins and IFN-inducible genes, and an improvement in cerebral blood flow. Several other novel treatments are being explored for AGS, as well as SLE, including inhibitors of cGAS [173], and tyrosine kinase 2 (TYK2), which mediates IFN receptor signaling. Notably, the suppression of inflammation mediated by type I IFNs (potentially triggered by L1 DNA) is a common theme among these potential SLE therapies.

Based on the biology of L1 and HERVs, agents that promote genomic CpG island methylation or other suppressive epigenetic events, or that prevent the translation of their transcripts, e.g., siRNAs, could also be developed for a more uniquely targeted treatment of SLE. The testing of such agents would also go a long way to validate, or refute, the pathogenic relevance of retrotransposons. Lastly, to the best of our knowledge, there is nothing in the drug development pipeline specifically for type I IFN-independent SLE, which mechanistically remains an enigma.

## 4. Concluding Remarks

The very modest successes in SLE drug development in modern times, and the shortcomings of mainstream models for its etiopathogenesis, make it apparent that new ideas are needed. A more reliable early diagnosis, more accurate prognostication, and the development of more effective treatments with better safety profiles, are all highly needed. To this end, the emerging evidence of endogenous retroelement involvement in SLE offers a tantalizing promise of progress.

While a broader set of retrotransposons may have varying degrees of involvement in initiating and perpetuating SLE and its flares, current evidence suggests that the L1 retrotransposon is likely the most consequential. However, a true causative role will need to be demonstrated by clinical trials using drugs that interfere with relevant aspects of L1 biology, e.g., reverse transcriptase inhibitors.

## Figures and Tables

**Figure 1 jcm-10-00856-f001:**
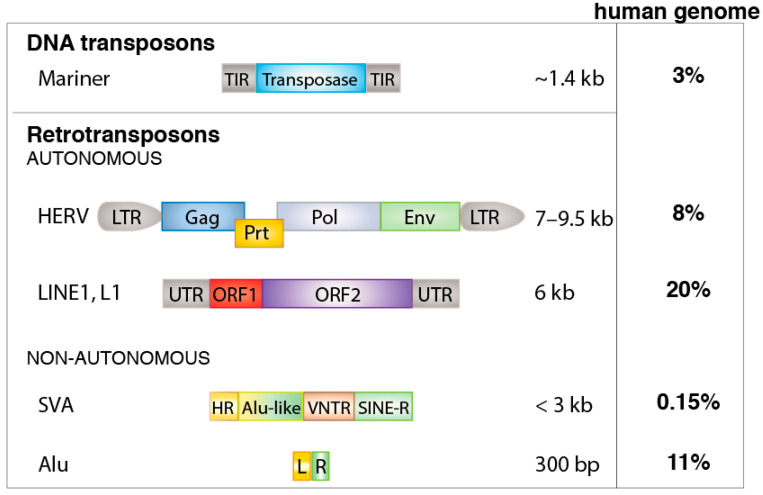
Classes and examples of transposable elements. Abbreviations: TIR, terminal inverted repeats; LTR, long terminal repeats (transcriptional control sequence); Gag, group antigen; Prt, protease; Pol, polymerase; Env, envelope; UTR, untranslated region; ORF, open reading frame; HR, hexamer repeat; VNTR, variable number tandem repeats; SINE-R, Alu right monomer; SVA, SINE-R/VNTR/Alu composite; L, left monomer; R, right monomer.

**Figure 2 jcm-10-00856-f002:**
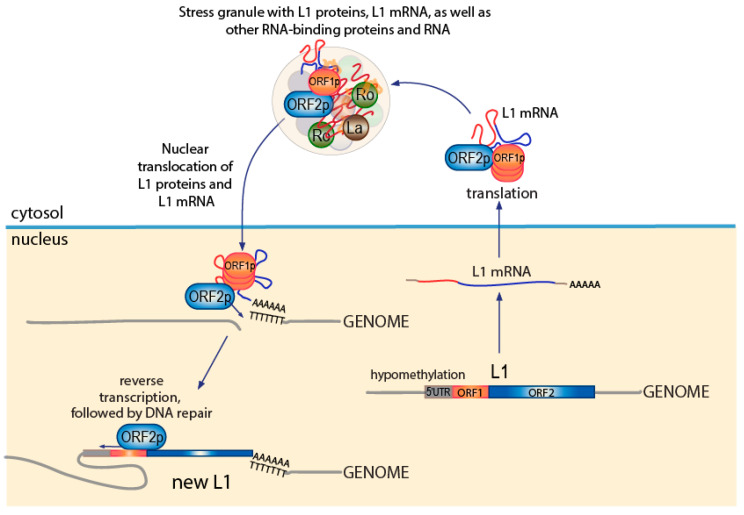
The L1 retrotransposition process. A similar figure is in [40].

**Figure 3 jcm-10-00856-f003:**
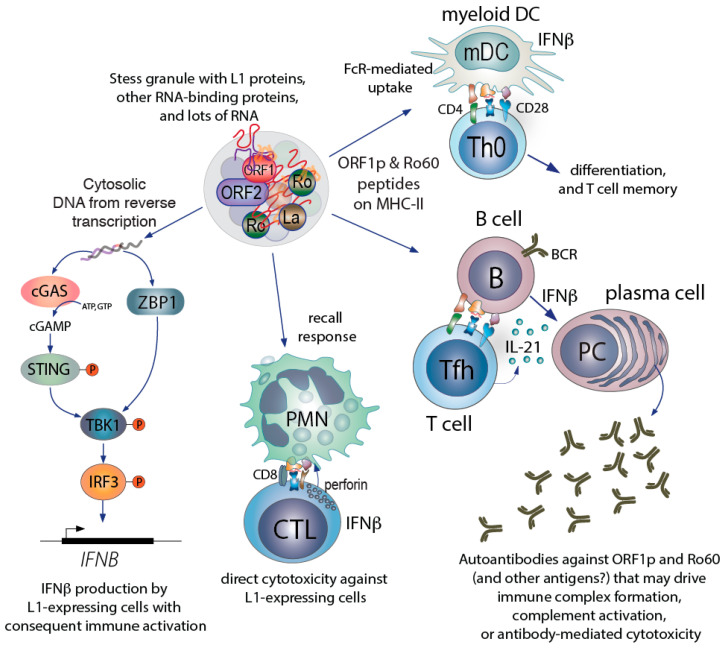
How the L1-containing stress granule may participate in driving SLE-related autoimmunity. The initial priming of T cells likely occurs by dendritic cells, which can take up stress granules, process their contents, and present peptides on class II MHC. The production of autoantibodies against ORF1p and Ro60 likely involves T cells primed by DC, followed by differentiation into follicular or peripheral helper T cells. CD8 T cells derived by cross-priming likely can recognized L1-expressing cells by virtue of ORF1p (and Ro60) derived peptides on class I MHC. Intracellularly, the reverse transcription of L1 transcripts into DNA will trigger IFNβ production and secretion. The secreted IFNβ will further stimulate monocyte differentiation to myeloid DC, plasma cell differentiation, and the differentiation and activation of CD8 T cells to become cytotoxic. Some cells do not express cGAS, but instead have other DNA sensors, such as Z-DNA binding protein 1, ZBP1, which also induce IFNβ production. Lastly (and not specifically illustrated), immune complexes that contain ORF1p, Ro60, and RNA (i.e., stress granules) will be taken up by plasmacytoid dendritic cells and B cells to trigger TLR-mediated IFNα production.

## Data Availability

Not applicable.

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
