# Peer review of "Implications of Endogenous Retroelements in the Etiopathogenesis of Systemic Lupus Erythematosus"

_jcm, 2021, doi:10.3390/jcm10040856_

Round 1

Reviewer 1 Report

This is a brilliant and useful overview on the potential implications of endogenous retroelements in the pathogenesis of SLE. The Authors presented data very clearly (figures are explanatory) and depicted their perspective providing the readerships with both the favouring elements and potential pitfalls.

I really appreciated reading this review.

Author Response

We thank the reviewer for a very positive review!

Reviewer 2 Report

The authors review the evidence for retroelements in the pathogenesis of lupus. There are other reviews on this topic, nevertheless, this is an active area and an important hypothesis and merits constant updating. The review is reasonably balanced but some attention could be given to the following:

  1. LINE-1 demethylation has not been shown to be different in lupus and controls in all studies. Some discuss of the possible mechanisms and implications of these discrepancies would be helpful
  2. Carter et al reported p40 LINE-1 antibodies in all tested individuals (not just lupus patients) but detected an increased titre in lupus when assayed by ELISA. Crow in correspondence for this paper raised the reasonable suggestion that the purported p40 antibodies might be binding to DNA. The authors might like to comment on this possibility and provide some clarification.
  3. As the authors state in their introductory remarks, a striking and still unresolved aspect of lupus is the sexual dimorphism. What can the retroelement hypothesis contribute to this question. In particular, what of the role of retroelements on X chromosome, and defects in X-inactivation?
  4. There is evidence that hypomethylation varies in different cell types, and indeed in different cells within a lineage. Is this important, and can the authors comment on whether expression is specific cell types (e.g. plasmacytoid dendritis cells, which make abundant type I IFN) is more important for development of SLE.
  5. The authros should include and discuss the Alu link with Ro60 and the IFNa axis as described by Hung et al, 2015
  6. It would be appropriate to cite Stetson et al, Cell, 2008 regarding the importance of TREX1 for removing retroelements and offsetting the risk of autoreactivity, as this was a seminal finding for the field.
  7. The model in figure 3 requires greater exposition if it is really to be taken to account for most features of SLE. At present, it is unclear which antigens are proposed to account for the cognate interaction between B cell and Tfh on the right of the figure, and how this relates to the stress granule.

Author Response

We thank this reviewer for the thorough review and excellent suggestions for improvement, which we follow to the extent that published data can address the questions that the reviewer poses. In detail:

  1. Yes, we agree. However, given that ~500,000 L1 elements exist throughout the human genome, a global L1 methylation analysis is not necessarily very informative. The key question is what the methylation status is of those L1 elements that are 1) transcriptionally activated in SLE, and 2) functional and full-length. We are investigating this, but have not yet published any data.
  2. Yes, we now mention these important controls, which the reviewer of that paper (Dr. Crow) indeed asked for.
  3. This is an important question, for which no definitive information yet exists. We mention that both HERVs and L1 may be transcriptionally stimulated by female hormones, but this has only been published for HERV-K. The X-chromosome contains many retroelements (as do all other chromosomes), but we are not aware of any data to show that they are expressed in autoimmunity. Our own unpublished data does not include transcripts from the X chromosome.
  4. We have some data on the expression of L1 in immune cell lineages, but it has not yet been published.
  5. We have added the Hung paper on Alu and Ro60. 
  6. We apologize for the omission of our colleague Dan Stetson's paper. We usually reference this key paper.
  7. The model was not meant to represent all of SLE, but merely the parts of the autoimmune response to L1 proteins and the proteins associated with them. While we are reluctant to provide more details about hypothetical aspects, we have made some refinements to the figure.

Reviewer 3 Report

The presented by Ukadike KC and Mustelin T manuscript review the role of endogenous retroelements in etiopathogenesis of autoimmune disease SLE. Recently, the authors were able to detect the anti-p40 antibodies at elevated levels in SLE patients, the finding that was supported by other groups working on L1. This fact together with the elevated expression of L1 transcripts in lupus patients makes LINE-1 elements strong contributors to induced and sustained autoimmune reactions.

While the manuscript is clear in describing the role of L1 and discussing the hypothesis recently proposed by the authors, there are some issues that require special attention.

Major comments:

  1. Following the logic of the Figure 1, it would be beneficial for the manuscript to review the role of Alu and SVA elements also in a separate chapter. In the presented manuscript only HERVs and L1 effects were presented. While, Alu repeats, for example, were reported to alter gene functions which results in various diseases, including autoimmune disorders. Therefore, the manuscript will be more complete in terms of reviewing the role of endogenous retroelements (all of them), as stated in the title, but not only LINEs-1 and HERVs.

  1. Lines 217-223. In summarizing the role of L1 in SLE, the authors are very categorical in assigning the major role in every aspect and symptoms of SLE to L1 activation. While, the possible role in triggering autoimmunity by L1 is now becoming more evident, I would discourage authors from making the L1 the main culprit in SLE pathogenesis.    

  1. The reference list needs very serious update.
  • It contains reiterations (for example, ref 37 and 71 is the same publication: Mustelin, T.; Ukadike, K.C. How Retroviruses and Retrotransposons in Our Genome May Contribute to Autoimmunity in 529 Rheumatological Conditions. Front Immunol 2020, 11, 593891, doi:10.3389/fimmu.2020.5938).
  • Several references should be replaced with the original publications rather than reviews, e.g., Ref24 on PD-1 gene should refer to doi: 10.1038/ng1020; Ref25 – on BANK1 gene to doi: 10.1038/ng.79.
  • Self-referencing to another review by the same corresponding author is unacceptable, e.g.: Mustelin T, 2019-doi: 10.3389/fimmu.2019.01028, Mustelin T, 2020-doi:10.3389/fimmu.2020.593891. Moreover, Figure 2 of (doi:10.3389/fimmu.2020.593891) greatly resembles the Figure 2 of the current manuscript.

  1. Lines 171-182 in the paragraph about L1 and methylation should also mention the publication by Mavragani et al. ,2018 (doi.org/10.1016/j.jaut.2017.10.004), which describes the links between SLE-associated MeCP2 gene and hypomethylation of L1. There is a growing body of evidence indicating that genetic association signals for SLE and other autoimmune diseases located within the genes whose products perform various modifications of genomic DNA thereby re-activating silent genes and repetitive elements, like LINE-1. The manuscript will only benefit if a chapter describing genetic links between relevant SLE-susceptibility genes and L1 will be added. This will provide an answer to the author’s question (lines 246-247): “Why and how does L1 become dysregulated in individuals who develop SLE? ”.

Minor comment:

Abstract, line 22. “can be largely explained by this model” is an overstatement to my mind. Perhaps, “partially, or to some extent” would be more correct to say.

Author Response

We thank the reviewer for the throughful review and the constructive suggestions, which we have followed as much as possible. In detail:

  1. We have added a section on Alu and SVA elements, which depend entirely on the retrotransposition L1 machinery for their genesis. 
  2. We apologize for the impression that we assign all aspects of SLE to L1. This was not our intent and we now soften the statements.
  3. We have made the recommended amendments and additions to the reference list. We do feel that it is appropriate to cite new ideas that we published in previous reviews, rather than proposing them as new ideas again.
  4. We now cite the Mavgrani paper. We have renewed our efforts to find genetic connections to L1 biology among the genes that are associated with SLE. 

minor: we have softened the statement as suggested.

Round 2

Reviewer 2 Report

Congratulations on an informative review

Author Response

Thank you very much!

Reviewer 3 Report

The authors did not comment on the Figure 2 similarity with another Figure from the last-year review by the authors, and haven't modified the abstract (line 22), as previously suggested (minor comment).

Author Response

We apologize for overlooking the request for line 22 in the abstract; this has now been remedied.

Figure 2 is indeed the creation of the same artist (=the senior author), but is also different from our previous paper in a number of ways. The replication of the Alu element is not present in the Figure 2 and many texts are different. The nucleus is colored to better emphasize cytosolic vs nuclear events. There are no issues with copy right.